# Analysis of SARS-CoV-2 antibodies in COVID-19 convalescent blood using a coronavirus antigen microarray

Rafael R. de Assis[1], Aarti Jain[1], Rie Nakajima[1], Algis Jasinskas[1], Jiin Felgner[1], Joshua M. Obiero [1],
Philip J. Norris[2,3], Mars Stone[2,3], Graham Simmons[2,3], Anil Bagri[4], Johannes Irsch[4], Martin Schreiber[5],
Andreas Buser [6], Andreas Holbro[5], Manuel Battegay[6], Philip Hosimer[7], Charles Noesen[7],
Oluwasanmi Adenaiye[8], Sheldon Tai[8], Filbert Hong[8], Donald K. Milton [8], D. Huw Davies[1], Paul Contestable[7],
Laurence M. Corash[4], Michael P. Busch [2,3], Philip L. Felgner [1] & Saahir Khan [9✉]

The current practice for diagnosis of COVID-19, based on SARS-CoV-2 PCR testing of pharyngeal or respiratory specimens in a symptomatic patient at high epidemiologic risk, likely underestimates the true prevalence of infection. Serologic methods can more accurately estimate the disease burden by detecting infections missed by the limited testing performed to date. Here, we describe the validation of a coronavirus antigen microarray containing immunologically significant antigens from SARS-CoV-2, in addition to SARS-CoV, MERS-CoV, common human coronavirus strains, and other common respiratory viruses. A comparison of antibody profiles detected on the array from control sera collected prior to the SARS-CoV-2 pandemic versus convalescent blood specimens from virologically confirmed COVID-19 cases demonstrates near complete discrimination of these two groups, with improved performance from use of antigen combinations that include both spike protein and nucleoprotein. This array can be used as a diagnostic tool, as an epidemiologic tool to more accurately estimate the disease burden of COVID-19, and as a research tool to correlate antibody responses with clinical outcomes.

[1] Department of Physiology and Biophysics, School of Medicine, University of California, Irvine, CA, USA. [2] Vitalant Research Institute, San Francisco, CA, USA. [3] Department of Laboratory Medicine, University of California, San Francisco, CA, USA. [4] Cerus Corporation, Concord, CA, USA. [5] Department of Surgery, Oregon Health & Science University, Portland, OR, USA. [6] Division of Infectious Diseases & Hospital Epidemiology, University Hospital Basel, University of Basel, Basel, Switzerland. [7] Ortho Clinical Diagnostics, Rochester, NY, USA. [8] Institute for Applied Environmental Health, School of Public Health, University of Maryland, College Park, MD, USA. [9] Division of Infectious Diseases, Department of Medicine, Keck School of Medicine, University of Southern California, Los Angeles, CA, USA. ✉email: saahirkh@usc.edu

COVID-19 caused by the SARS-CoV-2 virus is a worldwide pandemic with significant morbidity and mortality estimates from 0.5 to 2% of confirmed cases[1]. The current case definition for confirmed SARS-CoV-2 infection relies on PCR-positive pharyngeal or respiratory specimens, with testing largely determined by presence of fever or respiratory symptoms in an individual at high epidemiologic risk. However, this case definition likely underestimates true prevalence, as individuals who develop subclinical infection that does not produce fever or respiratory symptoms are unlikely to be tested, and testing by PCR of pharyngeal or respiratory specimens is only around 60–80% sensitive depending on sampling location and technique and the patient's viral load[2]. Widespread testing within the United States is also limited by the lack of available testing kits and testing capacity limitations of available public and private laboratories. Therefore, the true prevalence of SARS-CoV-2 infection is likely much higher than currently reported case numbers would indicate.

Serology can play an important role in defining the true prevalence of COVID-19, particularly for subclinical infection[2]. Early studies of serology demonstrate high sensitivity to detect confirmed SARS-CoV-2 infection, with antibodies to virus detected ~1–2 weeks after symptom onset[3]. Unlike PCR positivity, SARS-CoV-2 antibodies are detectable throughout the disease course and persist beyond the symptomatic phase[4]. Multiple serologic tests have been developed for COVID-19[5]. However, these tests are limited to detection of antibodies against one or two antigens, and cross-reactivity with antibodies to other human coronaviruses that are present in all adults[6] is currently unknown, resulting in limitations in test performance[7]. Prior use of serology for detection of emerging coronaviruses focused on antibodies against the spike (S) protein, particularly the S1 domain, and the nucleocapsid protein (NP)[8]. However, the optimal combination of antigens to most accurately detect strain-specific coronavirus antibodies remains a subject of debate.

Protein microarray technology can be used to detect antibodies of multiple isotypes against hundreds of antigens in a high throughput manner[9,10] so is well suited to serologic surveillance studies. This technology, which has previously been applied to other emerging coronaviruses[11], is based on detection of binding antibodies, which are well-correlated with neutralizing antibodies[12] but do not require viral culture in biosafety level 3 facilities. Recently, our group developed a coronavirus antigen microarray (CoVAM) that includes antigens from SARS-CoV-2 and tested it on human sera collected prior to the pandemic to demonstrate low cross-reactivity with antibodies from human coronaviruses that cause the common cold, particularly for the S1 domain[2]. Compared to other recently developed SARS-CoV-2 antigen microarrays[13,14], the CoVAM described here includes more comprehensive coverage of seasonal human coronaviruses and more variants of the immunodominant spike protein. We describe the validation of this methodology using convalescent blood specimens from COVID-19 cases confirmed by positive SARS-CoV-2 PCR.

## Results

**Construction of coronavirus antigen microarray.** A coronavirus antigen microarray (COVAM) was constructed containing 61 antigens that are causes of acute respiratory infections, including 11 antigens from SARS-CoV-2. The viral antigens printed on this array are from epidemic coronaviruses including SARS-CoV-2, SARS-CoV, and MERS-CoV, common cold coronaviruses (HKU1, OC43, NL63, 229E), and multiple subtypes of influenza, adenovirus, metapneumovirus, parainfluenza, and respiratory syncytial virus as listed in Supplementary Table 2. The SARS-CoV-2 antigens on this array include spike protein (S), including the receptor-binding (RBD), S1, and S2 domains and the whole protein (S1 + S2), nucleocapsid protein (NP), and papain-like protease (PLpro). There is a similar set of antigens represented on the array from SARS-CoV, MERS-CoV, and the four common cold coronaviruses.

**Discrimination of SARS-CoV-2 convalescent blood specimens using coronavirus antigen microarray.** To determine the antibody profile of SARS-CoV-2 infection, the differential IgG and IgM reactivity to these antigens was evaluated for 100 SARS-CoV-2 convalescent blood specimens from PCR-positive individuals (positive group) and 88 sera collected prior to the COVID-19 pandemic from naive individuals (negative control group). As shown in the heatmap (Fig. 1), the positive group is highly reactive against SARS-CoV-2 antigens. The negative controls do not show high reactivity overall to SARS-CoV-2, SARS-CoV or MERS-CoV antigens despite showing high reactivity to the common cold coronavirus antigens. The discrimination of the positive and negative groups is much greater for IgG antibodies than for IgM antibodies, although the overall pattern of reactivity is similar. Both positive and negative groups included both serum and plasma, and no systematic differences were observed between antibody response detected in serum versus plasma specimens. The two groups do not differ significantly in reactivity to antigens from common cold coronaviruses or other respiratory viruses for either IgG or IgM.

With respect to specific antigens, the positive group displays high IgG reactivity to SARS-CoV-2 NP and S antigens, including RBD, S1, and S2 domains (Fig. 2). The positive group also demonstrates IgM and IgG cross-reactivity against NP and S2-containing antigens from other betacoronaviruses (e.g., SARS, MERS, HKU1, OC43), while the negative group demonstrates low-level cross-reactivity to these antigens from SARS-CoV-2, likely due to high sequence homology of NP and S2 antigens among betacoronaviruses.

As an additional control, an earlier iteration of the CoVAM was tested against a separate set of pre-pandemic sera of 144 individuals that were monitored for acute respiratory virus infections including with seasonal betacoronaviruses. These individuals showed similar antibody cross-reactivity against SARS-CoV-2 antigens as the negative group (Supplementary Fig. 1). The two iterations of the CoVAM were printed and probed separately but demonstrated similar results indicating assay reproducibility.

**Determination of optimal antigen combinations to detect SARS-CoV-2 infection.** The differences between the groups appear to be restricted to SARS-CoV-2 antigens, so these antigens were the focus of subsequent analysis. The performance of all SARS-CoV-2 antigens on the CoVAM was defined based on differential reactivity in positive and negative controls using Receiver Operating Characteristic (ROC) Area Under Curve (AUC) with optimal cutoff chosen to achieve high specificity (Tables 1 and 2). For a selection of high-performing antigens for IgG and IgM, the ROC curves and differential reactivities between positive and negative controls are shown in Supplementary Figs. 2, 3. Among individual antigens for detection of IgG and IgM, S1 and RBD antigens demonstrated higher performance overall than NP and S2 antigens, while PLpro demonstrated the lowest performance.

In order to estimate the gain in performance by combining antigens, all possible combinations of antigens were tested in silico for performance in discriminating the positive and negative groups. The ROC curve with AUC, sensitivity, and specificity was

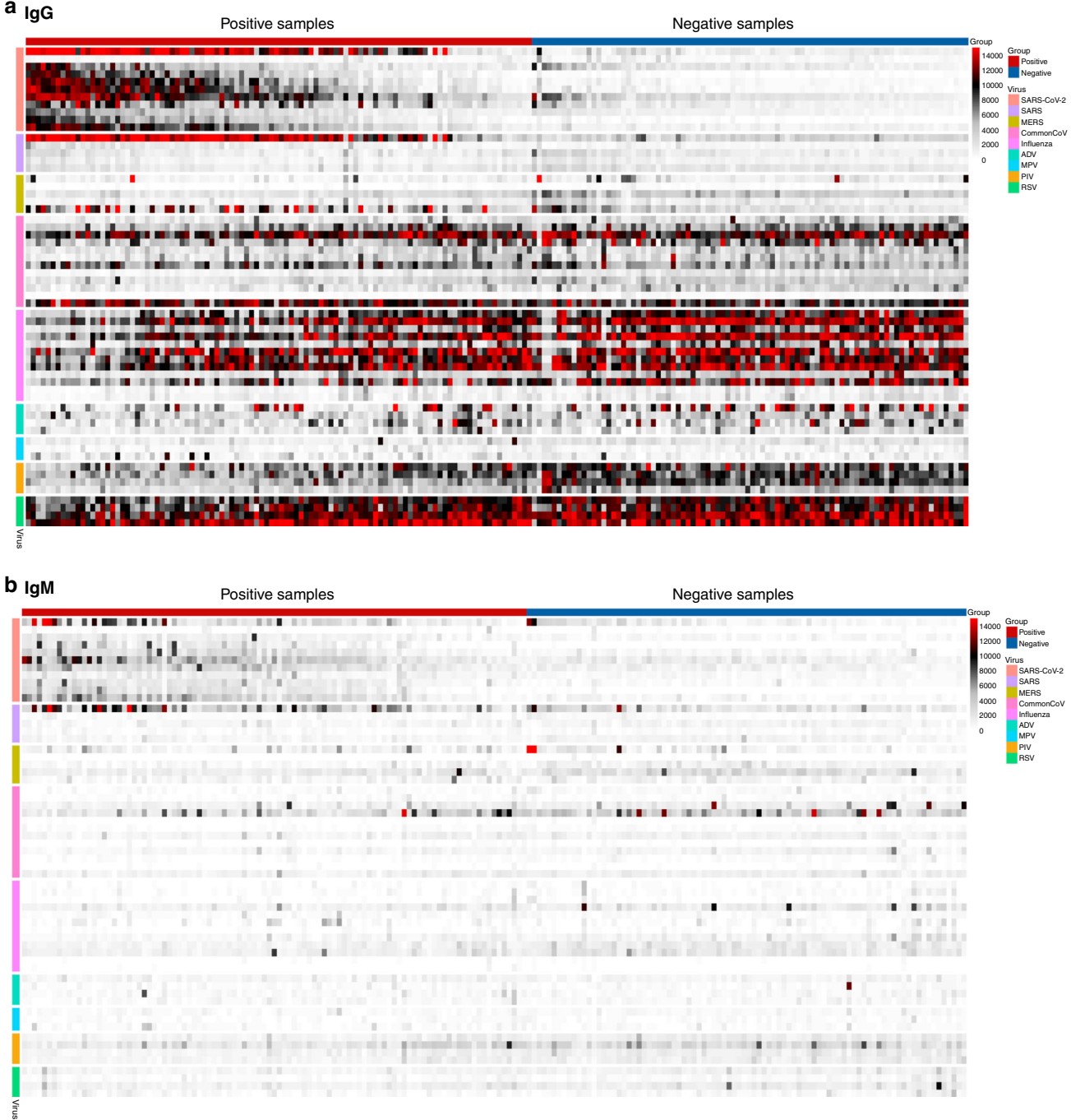

**Fig. 1 Heatmap for coronavirus antigen microarray.** The heatmap shows IgG (top) and IgM (bottom) reactivity measured as mean fluorescence intensity across four replicates, against each antigen organized into rows color coded by virus, for blood specimens organized into columns classified as positive (convalescent from PCR-positive individuals) or negative (prior to pandemic from naive individuals). Reactivity is represented by color (white = low, black = mid, red = high). Source data are provided as a Source data file.

calculated for each combination. The top-performing antigen combinations of each length are shown in Tables 3, 4, while complete data for all combinations is reported in Supplementary Data 1 in Supplementary information. For both IgG and IgM, there is a clear gain in performance by combining up to 4 antigens as shown in Fig. 3. However, for combinations of more than 4 antigens, test performance decreases due to decreasing specificity. The optimal antigen combinations include S1 and RBD antigens for both IgG and IgM but also include NP antigen for IgG and S2 antigen for IgM. These optimal combinations are not entirely predictable from the performance of individual antigens indicating that individual antigens are not orthogonal.

The performance of the assay can be further improved by combining IgG and IgM. For detection of prior SARS-CoV-2 infection at least 7 days post symptom onset, the combination of IgG and IgM achieves overall sensitivity 87.8% and specificity 98.9% for a high-specificity threshold and sensitivity 92.9% and specificity 97.7% for a high-sensitivity threshold.

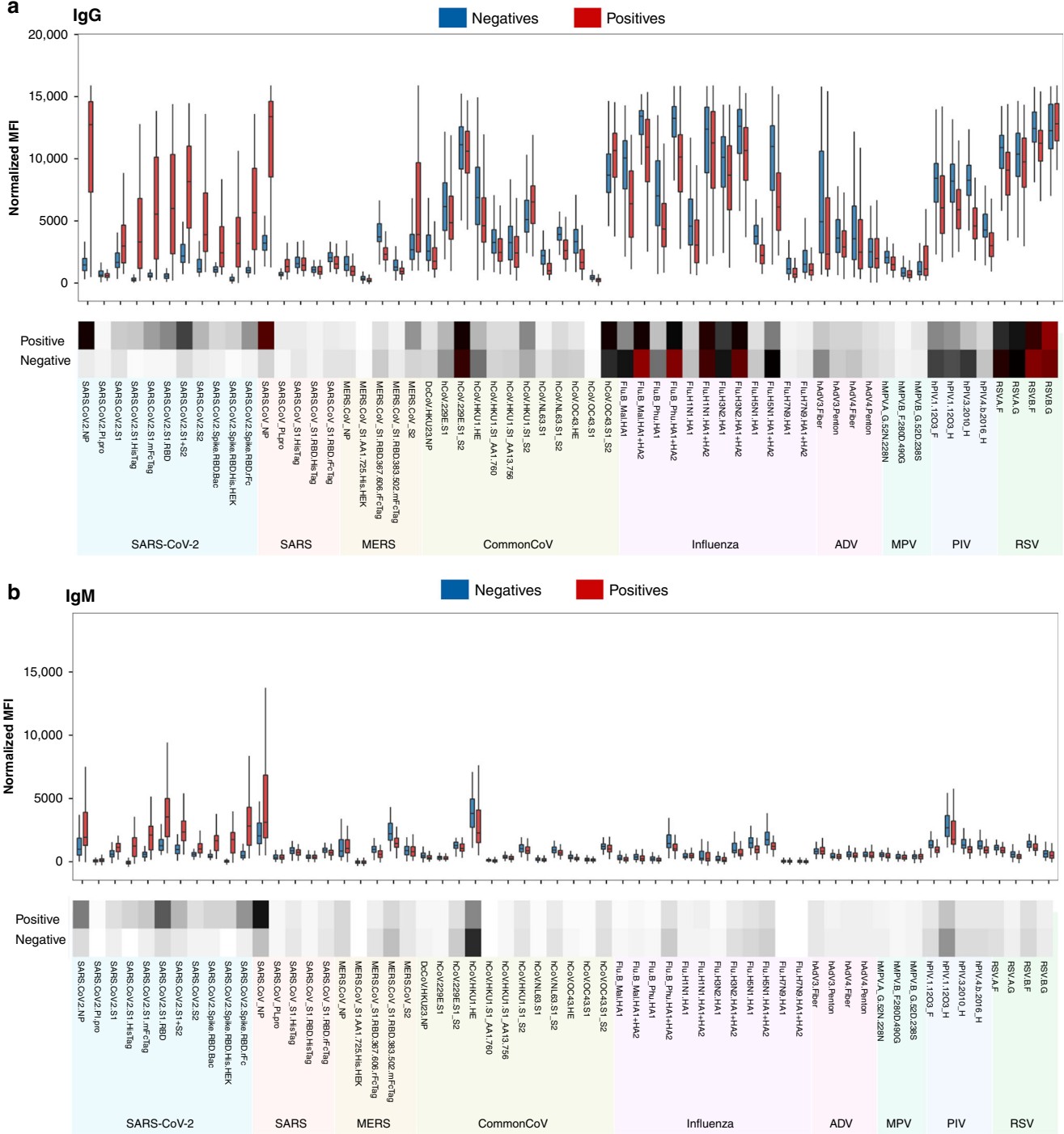

**Fig. 2 Normalized antibody reactivity of positive and negative sera on coronavirus antigen microarray.** The plot shows IgG (**a**) and IgM (**b**) reactivity against each antigen measured as mean fluorescence intensity (MFI). The boxes represent the first quartile, median and third quartile and the whiskers extend 1.5 times the interquartile range (IQR). Each point represents the distribution of the reactivity for either convalescent blood specimens from PCR-positive individuals (positive, red, n = 100) and sera from naive individuals prior to pandemic (negative, blue, n = 88). Below the plot, the heatmap shows average reactivity for each group (white = low, black = mid, red = high). The antigen labels are color coded for respiratory virus group. Source data are provided as a Source data file.

**Determination of SARS-CoV-2 antibody kinetics using CoVAM.** The CoVAM methodology based on the optimal antigen combinations identified above was used to measure the kinetics of the SARS-CoV-2 antibody response (Fig. 4). A total of 99 specimens collected longitudinally from PCR-positive individuals between

2 days and 32 days post symptom onset were used for this analysis. The earliest detection of IgM was at 5 days post symptom onset, and the earliest detection of IgG was at 7 days post symptom onset. Sensitivity for both IgG and IgM increased with time since symptom onset, and all individuals were positive for IgG by symptom day 22.

**Table 1 Ranking of SARS-CoV-2 antigens for IgG.**

| Rank | Antigen | Sensitivity | Specificity | AUC |
|---|---|---|---|---|
| **1** | **SARS.CoV.2.S1.mFcTag** | **0.5629** | **0.9849** | **0.9848** |
| 2 | SARS.CoV.2.S1.RBD | 0.6714 | 0.9845 | 0.9844 |
| 3 | SARS.CoV.2.NP | 0.6736 | 0.9843 | 0.9842 |
| 4 | SARS.CoV.2.S2 | 0.2193 | 0.9845 | 0.9842 |
| 5 | SARS.CoV.2.S1 | 0.0985 | 0.9847 | 0.9841 |
| 6 | SARS.CoV.2.S1+S2 | 0.2585 | 0.9843 | 0.9841 |
| 7 | SARS.CoV.2.Spike.RBD.Bac | 0.41 | 0.9843 | 0.984 |
| 8 | SARS.CoV.2.Spike.RBD.His.HEK | 0.6696 | 0.984 | 0.9839 |
| 9 | SARS.CoV.2.Spike.RBD.rFc | 0.6716 | 0.9836 | 0.9835 |
| 10 | SARS.CoV.2.S1.HisTag | 0.7089 | 0.983 | 0.9829 |
| 11 | SARS.CoV.2.Pl.pro | 0.0106 | 0.9708 | 0.9701 |

Test performance characteristics for discrimination of positive and negative sera were derived for each individual antigen using ROC analysis with high specificity threshold, with top-performing antigen in bold.

**Table 2 Ranking of SARS-CoV-2 antigens for IgM.**

| Rank | Antigen | Sensitivity | Specificity | AUC |
|---|---|---|---|---|
| **1** | **SARS.CoV.2.Spike.RBD.rFc** | **0.5894** | **0.9873** | **0.987** |
| 2 | SARS.CoV.2.S1.HisTag | 0.6238 | 0.9867 | 0.9866 |
| 3 | SARS.CoV.2.S1 | 0.2198 | 0.9858 | 0.9854 |
| 4 | SARS.CoV.2.S1.mFcTag | 0.1389 | 0.9855 | 0.9852 |
| 5 | SARS.CoV.2.S2 | 0.474 | 0.9851 | 0.9848 |
| 6 | SARS.CoV.2.Spike.RBD.His.HEK | 0.6915 | 0.9843 | 0.9842 |
| 7 | SARS.CoV.2.NP | 0.0595 | 0.9844 | 0.9839 |
| 8 | SARS.CoV.2.S1.RBD | 0.5929 | 0.9839 | 0.9836 |
| 9 | SARS.CoV.2.S1+S2 | 0.5704 | 0.9839 | 0.9836 |
| 10 | SARS.CoV.2.Spike.RBD.Bac | 0.2222 | 0.9829 | 0.9827 |
| 11 | SARS.CoV.2.Pl.pro | 0.0413 | 0.9817 | 0.9809 |

Test performance characteristics for discrimination of positive and negative sera were derived for each individual antigen using ROC analysis with high specificity threshold, with top-performing antigen in bold.

**Table 3 Ranking of high-performing SARS-CoV-2 antigen combinations for IgG.**

| N | Antigen combination | Sensitivity | Specificity | AUC |
|---|---|---|---|---|
| 1 | S1.mFcTag | 0.5629 | 0.9849 | 0.9848 |
| 2 | S1; S1.HisTag | 0.7132 | 0.9888 | 0.9887 |
| 3 | NP; S1; S1.mFcTag | 0.6056 | 0.9858 | 0.9857 |
| **4** | **NP; S1; S1.mFcTag; Spike.RBD.Bac** | **0.6004** | **0.9861** | **0.986** |
| 5 | NP; S1; S1.mFcTag; S1+S2; S2 | 0.7895 | 0.9835 | 0.9835 |
| 6 | NP; S1.mFcTag; S1.RBD; S1+S2; Spike.RBD.Bac; Spike.RBD.His.HEK | 0.8077 | 0.9816 | 0.9816 |
| 7 | NP; S1.HisTag; S1.mFcTag; S1.RBD; S1+S2; Spike.RBD.Bac; Spike.RBD.His.HEK | 0.8566 | 0.9779 | 0.9779 |
| 8 | NP; Pl.pro; S1; S1.HisTag; S1.mFcTag; S1+S2; Spike.RBD.Bac; Spike.RBD.rFc | 0.9147 | 0.9749 | 0.9749 |
| 9 | NP; Pl.pro; S1; S1.mFcTag; S1.RBD; S1+S2; Spike.RBD.Bac; Spike.RBD.His.HEK; Spike.RBD.rFc | 0.9169 | 0.9645 | 0.9645 |
| 10 | NP; Pl.pro; S1; S1.HisTag; S1.mFcTag; S1.RBD; S1+S2; Spike.RBD.Bac; Spike.RBD.His.HEK; Spike.RBD.rFc | 0.9128 | 0.9567 | 0.9567 |
| 11 | NP; Pl.pro; S1; S1.HisTag; S1.mFcTag; S1.RBD; S1+S2; S2; Spike.RBD.Bac; Spike.RBD.His.HEK; Spike.RBD.rFc | 0.918 | 0.9358 | 0.9359 |

Test performance characteristics for discrimination of positive and negative sera were derived for each high-performing antigen combination using ROC analysis with high specificity threshold, with optimal antigen combination in bold.

## Discussion

This study reveals several insights into the antibody response to SARS-CoV-2 infection with implications for vaccine development. The antibody profiles of naive individuals include high IgG reactivity to common cold coronaviruses with low-level cross-reactivity with NP and S2 domains from SARS-CoV-2 and other epidemic coronaviruses, which is not surprising given the high degree of sequence homology and previously observed serologic cross-reactivity[15] between NP and S2 domains of betacoronaviruses, a group that includes SARS-CoV-2, SARS-CoV, MERS, and common cold coronaviruses HKU1 and OC43. This low-level cross-reactivity occurs in approximately 7% of unexposed individuals (Fig. 1), which leads to hypotheses regarding whether these individuals differ in COVID-19 susceptibility and outcomes. The low-level antibody cross-reactivity of a subset of unexposed individuals for SARS-CoV-2 spike protein containing

**Table 4 Ranking of high-performing SARS-CoV-2 antigen combinations for IgM.**

| N | Antigen combination | Sensitivity | Specificity | AUC |
|---|---|---|---|---|
| 1 | Spike.RBD.rFc | 0.5894 | 0.9873 | 0.987 |
| 2 | S1; S1.HisTag | 0.6462 | 0.9865 | 0.9864 |
| 3 | S1.RBD; S1+S2; S2 | 0.7838 | 0.9876 | 0.9876 |
| **4** | **S1; S1.mFcTag; S1.RBD; S2** | **0.7714** | **0.9883** | **0.9883** |
| 5 | Pl.pro; S1; S1.mFcTag; S1.RBD; Spike.RBD.His.HEK | 0.7021 | 0.987 | 0.9868 |
| 6 | Pl.pro; S1; S1.mFcTag; S1.RBD; S1+S2; Spike.RBD.His.HEK | 0.748 | 0.9855 | 0.9854 |
| 7 | NP; S1; S1.mFcTag; S1.RBD; S1+S2; Spike.RBD.His.HEK; Spike.RBD.rFc | 0.7519 | 0.9842 | 0.9841 |
| 8 | Pl.pro; S1; S1.mFcTag; S1.RBD; S1+S2; Spike.RBD.Bac; Spike.RBD.His.HEK; Spike.RBD.rFc | 0.7647 | 0.981 | 0.9809 |
| 9 | S1; S1.HisTag; S1.mFcTag; S1.RBD; S1+S2; S2; Spike.RBD.Bac; Spike.RBD.His.HEK; Spike.RBD.rFc | 0.7904 | 0.9757 | 0.9757 |
| 10 | NP; S1; S1.HisTag; S1.mFcTag; S1.RBD; S1+S2; S2; Spike.RBD.Bac; Spike.RBD.His.HEK; Spike.RBD.rFc | 0.7963 | 0.9675 | 0.9675 |
| 11 | NP; Pl.pro; S1; S1.HisTag; S1.mFcTag; S1.RBD; S1+S2; S2; Spike.RBD.Bac; Spike.RBD.His.HEK; Spike.RBD.rFc | 0.7958 | 0.961 | 0.961 |

Test performance characteristics for discrimination of positive and negative sera were derived for each high-performing antigen combination using ROC analysis with high specificity threshold, with optimal antigen combination in bold.

**Fig. 3 ROC curves for high-performing combination of antigens.** ROC curves showing sensitivity versus specificity for discrimination of positive and negative blood specimens were derived for each combination of the high performing antigens for both IgM (**a**) and IgG (**b**) and compared to no discrimination (ROC AUC = 0.5, gray line). Source data are provided as a Source data File.

S2 domain may influence response to vaccination with spike protein antigens containing the S2 domain in this subset of individuals.

This study also informs antigen selection and design for population surveillance and clinical diagnostic assays. The optimal binding antibody assay to discriminate SARS-CoV-2 convalescent sera from pre-pandemic sera is a combination of 4 antigens that include S1, RBD, and either S2 or NP. As individual antigens, NP and S2 demonstrate cross-reactivity with negative control sera which leads to lower specificity, but these antigens add predictive power when combined with the more specific S1 and RBD antigens. Larger combinations of antigens actually show decreased predictive power due to decreased specificity. In addition, differences among variants with different purification tags indicate that these tags may have a significant effect on the antigen conformation resulting in differences in antibody binding.

The overall diagnostic performance of this CoVAM is within range of existing serodiagnostic tests for SARS-CoV-2[7]. However, most existing serodiagnostic tests for SARS-CoV-2 do not report validation data as early as 7 days post symptom onset, and when examining only specimens collected at least 14 days post symptom onset, CoVAM performs better than individual antigen assays. Future variants of the CoVAM can include additional antigens that have shown some potential in selectively diagnosing coronavirus infections[16].

The coronavirus antigen microarray can be useful both as an epidemiologic tool and as a research tool. The high throughput detection of SARS-CoV-2-specific antibody profiles that reliably distinguish COVID-19 cases from negative controls can be applied to large-scale population surveillance studies for a more accurate estimation of the true prevalence of disease than can be achieved with symptom-based PCR testing. Recent development of more portable digital microscope technology that can be

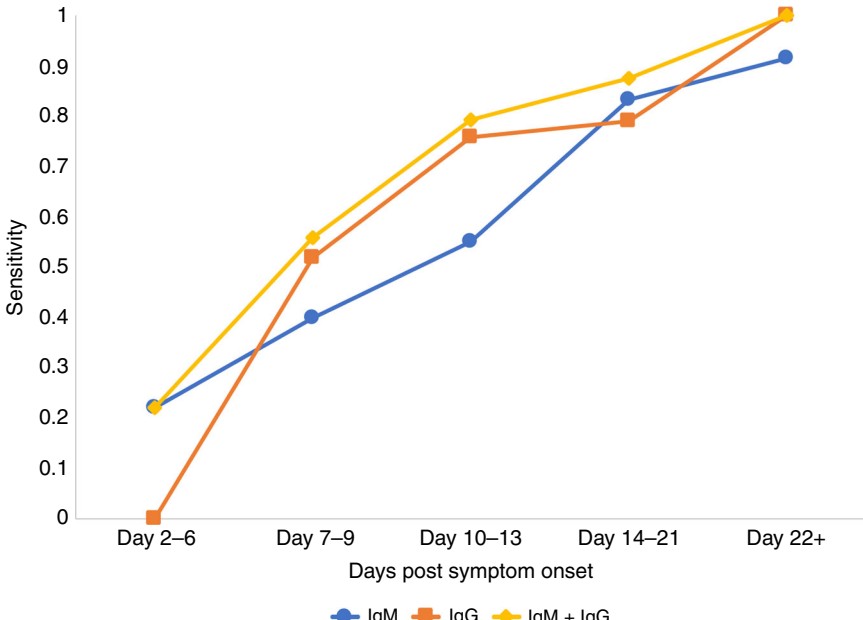

**Fig. 4 Sensitivity of CoVAM as a function of time since symptom onset.** The sensitivity in detecting SARS-CoV-2 antibody responses in PCR-positive individuals is shown as a function of time since symptom onset for IgM, IgG, and either IgM or IgG. Source data are provided as a Source data file.

integrated with smartphones will facilitate field implementation of the CoVAM. In addition, detection of these antibodies in SARS-CoV-2 convalescent plasma donations can provide validation prior to clinical use for passive immunization. The variation in the SARS-CoV-2 antibody profiles among acute and convalescent donors suggests that epitope characterization of convalescent donor plasma will be informative for evaluation of passive immune therapy efficacy in COVID-19 patients. The central role of inflammation in the pathogenesis of severe COVID-19[17] can be more closely studied by analyzing both strain-specific and cross-reactive antibody responses, particularly to test hypotheses regarding antibody-dependent enhancement with critical implications for vaccine development[18].

In conclusion, a coronavirus antigen microarray containing a panel of antigens from SARS-CoV-2 in addition to other human coronaviruses was able to reliably distinguish convalescent plasma of PCR-positive COVID-19 cases from negative control sera collected prior to the pandemic. Antigen combinations demonstrated improved performance compared to each individual antigen. Further studies are needed to apply this methodology to large-scale serologic surveillance studies and to correlate specific antibody responses with clinical outcomes.

## Methods

**Specimen collection**. A total of 135 de-identified SARS-CoV-2 convalescent blood specimens were collected from nasopharyngeal PCR-positive individuals from different sources with associated data on timing of symptom onset, positive PCR test, and collection (Supplementary Table 1). Of these convalescent specimens, 100 were collected from unique individuals at least 7 days post symptom onset so were used as positive controls, while 99 were collected longitudinally and used for the purpose of studying test performance as a function of time since symptom onset, with some overlap between the groups. A total of 88 de-identified pre-pandemic blood specimens were collected prior to November 1, 2019 from healthy blood donors and were used as negative controls. In addition, 144 specimens were collected between November 2018 and May 2019 from college students for a larger study of the contagious phenotype[19] and include specimens from individuals with acute respiratory virus infection, including seasonal coronaviruses. All biological specimens used in this study were collected under the auspices of local regulatory authorities governing protection of human research subjects with informed consent including future research use authorization.

**Specimen testing on coronavirus antigen microarray**. The coronavirus antigen microarray used in this investigation includes 60 antigens across respiratory virus subtypes, including 11 antigens from SARS-CoV-2 expressed in either baculovirus or HEK-293 cells (Supplementary Table 2). These antigens were provided by Sino Biological U.S. Inc. (Wayne, PA) as either catalog products or custom synthesis service products. The antigens were printed onto microarrays, probed with human sera, and analyzed as previously described[10,20,21].

Briefly, lyophilized antigens were reconstituted with sterile water to a concentration of 0.1 mg/mL bringing protein solution to 1× phosphate-buffered saline (PBS) and printing buffer was added. Antigens were then printed onto ONCYTE AVID nitrocellulose-coated slides (Grace Bio-Labs, Bend, OR) using an OmniGrid 100 microarray printer (GeneMachines). The microarray slides were probed with human sera diluted 1:100 in 1x Protein Array Blocking Buffer (GVS Life Sciences, Sanford, ME) overnight at 4 °C and washed with T-TBS buffer (20 mM Tris-HCl, 150 mM NaCl, 0.05% Tween-20 in ddH$_2$O adjusted to pH 7.5 and filtered) 3 times for 5 min each. A mixture of human IgG (Grace Bio-Labs SKU 110610) and IgM (Grace Bio-Labs SKU 110630), or IgG and IgA (Grace Bio-Labs SKU 110620), secondary antibodies conjugated to quantum dot fluorophores Q800 and Q585, respectively (antibodies obtained from Grace Bio-Labs, Bend, O) was applied to each of the microarray pads and incubated for 2 h at room temperature, and pads were then washed with T-TBS 3 times for 5 min each and dried. The slides were imaged using ArrayCam imager (Grace Bio-Labs, Bend, OR). Spot quantification (median spot fluorescence) and background signal subtraction was performed using the software ScanArray Express (PerkinElmer) version 3.0. Non-specific binding of secondary antibodies was subtracted using saline control. Mean fluorescence of the 4 replicate spots for each antigen was used for analysis.

**Statistics and reproducibility**. The mean fluorescence intensity (MFI) of each antigen was determined by the average of the median fluorescence signal of four replicate spots. The fluorescence signal for each spot was determined by its signal intensity subtracted by the background fluorescence. All statistical analyses were conducted using R version 4.0.0 (R Foundation for Statistical Computing, Vienna, Austria). All analysis and figures are representative of a single probing experiment.

MFI was normalized by the quantile normalization method using the *normalize. quantiles.use.target* function from the *preprocessCore* package (version 1.50.0). As a reference for normalization, a vector containing the median MFI for IgG or IgM was constructed. Descriptive statistics were used to summarize the IgM and IgG reactivity measured as MFI. Wilcoxon tests with $p < 0.05$ were used to compare the mean differences between groups.

In order to rank the individual antigens and combinations of antigens from SARS-CoV-2 for performance in discriminating the positive and negative groups, the receiver operating characteristic area under the curve (ROC AUC) values as well as sensitivity and specificity (derived from a logistic regression model) values for each antigen or combination of antigens were calculated by comparing positive and negative specimens using the *pROC* package (version 1.16.2) and the *stats* package (version 4.0.0). For this, the samples were randomly partitioned into two groups, at a ratio of 70%/30%, using the *caret* package (version 6.0-86). The group

with 70% (training set) of the samples was used to create a logistic regression model using the *glm* function form the *stats* package (version 4.0.0). The 30% (testing set) subset was used to predict the outcome of each sample being classified as negative or positive using the *stat* package and the performance (sensitivity and specificity) calculated. This process was repeated 500 times and the final sensitivity, specificity, and AUC values were calculated as the mean values of all repetitions.

All individual antigens and combinations of antigen were ranked based on the ROC AUC analysis above, and the top-performing combinations were further tested using the full sample set. The coordinates of the ROC curve were obtained (*coords* function from the pROC package) and the candidate combinations were compared to select the highest sensitivity at specificity equal to 1. Data visualization was performed using the *ggplot2* package (version 3.3.0) or *pROC* package.

**Reporting summary**. Further information on research design is available in the Nature Research Reporting Summary linked to this article.

## Data availability
The raw data that support the findings of this study are available from the corresponding author upon reasonable request. Source data are provided with this paper.

## Code availability
The complete custom R scripts used for data analysis are publicly available at the GitHub repository: https://doi.org/10.5281/zenodo.4146101.

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

## Acknowledgements
S. Khan is supported by the National Center for Research Resources and the National Center for Advancing Translational Sciences, National Institutes of Health, through Grant KL2 TR001416. The content is solely the responsibility of the authors and does not necessarily represent the official views of the NIH. Prometheus-UMD was sponsored by the Defense Advanced Research Projects Agency (DARPA) BTO under the auspices of Col. Matthew Hepburn through agreements N66001-17-2-4023 and N66001-18-2-4015 (PI: D. Milton). This project has been funded in part with federal funds from the Office of the Assistant Secretary for Preparedness and Response, Biomedical Advanced Research and Development Authority under agreement N6601-18-2-4015. This study was funded in part by the Defense Threat Reduction Agency via grants HDTRA1-18-1-0036 (PI: H. Davies) and HDTRA1-18-1-0035 (PI: P. Felgner). The findings and conclusions in this report are those of the authors and do not necessarily represent the official position or policy of the funding agencies and no official endorsements should be inferred. The Prometheus-UMD consortium includes the following investigators in alphabetical order: Addo, Kofi; Adenaiye, Oluwasanmi Oladapo; Agrawala, Agrawala; Aiello, Allison; Albert, Barbara; Arria, Amelia; Bueno de Mesquita, P. Jacob; Cai, Mara; Chen, Shuo; Chen, Wilbur; Corrada Bravo, Hector; Elworth, Leo; Felgner, Philip; Frieman, Matthew; German, Jennifer; Heidarinejad, Mohammad; Hong, Filbert; Jiang, Chengsheng; Khan, Saahir; Lai, Jianyu; Liu, Hongjie; Ma, Tianzhou; Maljkovic Berry, Irina; Martinello, Richard; Mattise, Nick; Memon, Atif; Milton, Donald; Mongodin, Emmanuel; Nasko, Dan; Pop, Mihai; Porter, Adam; Romo, Sebastian; Srebric, Jelena; Tai, Sheldon; Treangen, Todd; Wajid, Faizan; Washington-Lewis, Rhonda; Wu, Qiong; Xing, Yishi; Youssefi, Somayeh; Zhu, Shengwei.

## Author contributions
The coronavirus antigen microarray was designed by S. Khan and P. Felgner and was constructed by R. Nakajima and A. Jasinskas at UCI. The positive COVID-19 convalescent blood specimens and negative control specimens were collected as research specimens by L. Corash, A. Bagri, and J. Irsch at Cerus; M. Busch, G. Simmons, M. Stone, and P. Norris at Vitalant; M. Schreiber at OHSU; P. Hosimer, C. Noesen, and P. Contestable at Ortho Clinical Diagnostics; and A. Buser, A. Holbro, and M. Battegay at University Hospital Basel. A set of negative control sera used to validate an earlier iteration of the CoVAM were collected as research specimens for the Prometheus-UMD consortium by D. Milton, O. Adenaiye, S. Tai, and F. Hong at UMD. The testing of specimens on the coronavirus antigen microarray was performed by A. Jain and J. Felgner at UCI. The data analysis was performed by R. de Assis and J. Obiero at UCI. The manuscript and figures were prepared by S. Khan and R. de Assis with input and approval from all other authors.

## Competing interests
The coronavirus antigen microarray is the intellectual property of the Regents of the University of California that is licensed for commercialization to Nanommune Inc. (Irvine, CA), a private company for which P. Felgner is the largest shareholder and several co-authors (R. de Assis, A. Jain, R. Nakajima, A. Jasinskas, J. Obiero, Jiin Felgner, H. Davies, and S. Khan) also own shares. Nanommune Inc. has a business partnership with Sino Biological Inc. (Beijing, China) which expressed and purified the antigens used in this study. The convalescent plasma used in this study was collected for clinical use by independent blood centers using licensed plasma or platelet processing systems manufactured by Cerus Corporation, for which multiple authors (L. Corash, A. Bagri, and Johannes Irsch) are shareholders and employees. Convalescent sera were also provided by Ortho Clinical Diagnostics, which is using these specimens to validate a clinical diagnostic test, and P. Hosimer, C. Noesen, and P. Contestable are shareholders and employees of this company. M. Battegay, A. Buser, A. Holbro, Philip J. Norris, Mars Stone, Graham Simmons, Martin Schreiber, Oluwasanmi Adenaiye, Sheldon Tai, Filbert Hong, Donald K. Milton, Michael P. Busch, and the Prometheus-UMD consortium investigators have no conflicts of interest to disclose.
