## [Peer Review File · Nature Communications]

Reviewers' Comments:

Reviewer #1:

Remarks to the Author:

Strength

1. This is the first retrospective antibody study on the use of antigen microarray for the diagnosis of COVID-19.
2. The paper is well written.

Weakness

1. There is no comparison with existing ELISA test at all.
2. There is no study on the timing of first positivity, whether it is better or worse.
3. There is no field validation in realtime to show its superiority to differentiate COVID-19 and other acute respiratory tract infection.

Reviewer #2:

Remarks to the Author:

This manuscript by De Assis et al. developed a microarray to analyze the SARS-CoV-2 antibodies in the COVID-19 convalescent blood. The topic is of significant importance, especially during the current COVID-19 pandemic, and the manuscript is well written. Below are my comments:

1. In this work, S1, S2, full spike (S1+S2), NP and RBD were used to to build the microarray. It was unclear why some proteins were chosen while the others (e.g., membrane protein or envelop protein) were not selected when other article (<https://pubmed.ncbi.nlm.nih.gov/16081901/>) suggested that membrane protein can potentially be used for diagnosis for coronavirus. Please provide a rationale for why other proteins were not selected in either the introduction or methodology section.
2. In line 226, the results indicated that SARS-CoV-2 S1 tagged with mFc displayed much higher IgG/IgA reactivity. Can authors provide an explanation?
3. In line 296, authors stated that n protein typically showed high sequence homology among beta coronaviruses, yet the cross-reactivity were only shown between SARS-CoV and SARS-CoV-2. Please elaborate on why cross-reactivity was not shown in the discussion section.

Reviewer #3:

Remarks to the Author:

Dear Editor-In-Chief,

The manuscript, titled "Analysis of SARS-CoV-2 Antibodies in COVID-19 Convalescent Blood using a Coronavirus Antigen Microarray" by de Assis et al., describes the fabrication and application of a coronavirus protein array in diagnosis of SARS-CoV-2. First, the authors used commercially available S (including S1, S2, and RBD) and N proteins from SARS-CoV-2, SARS-CoV, MERS-CoV, and four common human coronavirus strains to construct their protein arrays. Next, they tested 22 de-identified SARS-CoV-2 convalescent blood specimens on their arrays and employed 144 de-identified pre-pandemic control sera as negative controls. Using ROC analysis, they demonstrated that both CoV-2 S and N proteins individually or in combination could discriminate the COVID-positives from the negative controls.

Although this study aimed to develop a diagnostic tool for COVID-19 pandemic, I have several major concerns. First, the size of the CoV-2 cohort is very small to make a statistically reliable

conclusion for the diagnosis. Comprehensive statistic analysis is needed. Second, only convalescent samples were tested in this study. As shown by many other studies, the SARS-CoV-2 specific antibody titre is a lot higher at the late stage. Sample of early stage need to be tested to demonstrate the sensitivity and usefulness of the identified biomarkers and biomarker panels. Third, I was hoping to see more proteins of SARS-CoV-2 were included on their arrays; however, only the S and N proteins obtained from a commercial resource were used. S (including S1, S2 and RBD) and N proteins have already been widely applied for diagnosis of COVID-19; many tests on the basis of S and N proteins have already been approved by the FDA. Thus, what value this study can add is unclear to me.

Minor points:

- 1) The authors should not ignore other highly related studies. More comprehensive SARS-CoV-2 protein microarrays have already been reported by other groups. (doi: <https://doi.org/10.1101/2020.03.20.20039495>, doi: <https://doi.org/10.1101/2020.03.26.994756>)
- 2) It was unclear why the authors tested anti-IgA rather than anti-IgM.
- 3) To assure the reliability of the array results, it is critical to assure the reproducibility of the newly fabricated microarray at the first instance
- 4) Because of the requirement of sophisticate expertise and special instruments, it is impractical to use protein microarray as a tool for routine diagnostics.

September 14, 2020

Dear referees,

Thank you for your valuable feedback in response to our initial submission. We have made major revisions to the manuscript, including generating a new version of the microarray containing additional antigens from SARS-CoV-2 and testing for IgG and IgM against a much larger set of confirmed case specimens that became available in the interim period. We believe the updated manuscript is significantly improved as a result of your comments. Please see our details responses below.

Sincerely,

Saahir Khan, MD, PhD
Assistant Clinical Professor, Infectious Diseases
2020 Zonal Ave. Rm. 433
University of Southern California
(650) 269-9466
saahirkh@usc.edu

Response to Reviewer 1:

Comment	Response
"There is no comparison with existing ELISA test at all."	We have included references to prior data comparing protein microarray to ELISA as well as comparative performance of ELISA tests approved to date.
"There is no study on the timing of first positivity, whether it is better or worse."	We have included data on test performance as a function of time relative to symptom onset in Figure 4.
"There is no field validation in realtime to show its superiority to differentiate COVID-19 and other acute respiratory tract infection."	We have included data from a pre-pandemic sera set that includes other acute respiratory infections in Supplementary Figure 1.

Response to Reviewer 2:

Comment	Response
"In this work, S1, S2, full spike (S1+S2), NP and RBD were used to to build the microarray. It was unclear why some proteins were chosen while the others (e.g., membrane protein or envelop protein) were not selected when other article suggested that membrane protein can potentially be used for diagnosis for coronavirus. Please provide a rationale for why other proteins were not selected in either the introduction or methodology section."	We have included as background the reasons for choosing NP and S proteins as the focus of the microarray, but we acknowledge in the discussion that inclusion of other proteins could improve performance based on new literature.
"In line 226, the results indicated that SARS-CoV-2 S1 tagged with mFc displayed much higher IgG/IgA reactivity. Can authors provide an explanation?"	We have included in the discussion a statement that different tags may influence the structural integrity of the proteins and thus impact performance.

“In line 296, authors stated that n protein typically showed high sequence homology among beta coronaviruses, yet the cross-reactivity were only shown between SARS-CoV and SARS-CoV-2. Please elaborate on why cross-reactivity was not shown in the discussion section.”	Inclusion of more diverse negative specimens in this updated manuscript demonstrates some increased cross-reactivity of NP protein.
---	--

Response to Reviewer 3:

Comment	Response
“The size of the CoV-2 cohort is very small to make a statistically reliable conclusion for the diagnosis. Comprehensive statistic analysis is needed.”	We have increased the sample size of the positive control specimens in this updated manuscript.
“Only convalescent samples were tested in this study. As shown by many other studies, the SARS-CoV-2 specific antibody titre is a lot higher at the late stage. Sample of early stage need to be tested to demonstrate the sensitivity and usefulness of the identified biomarkers and biomarker panels.”	We have included data on test performance relative to timing of symptom onset in Figure 4.
“I was hoping to see more proteins of SARS-CoV-2 were included on their arrays; however, only the S and N proteins obtained from a commercial resource were used. S (including S1, S2 and RBD) and N proteins have already been widely applied for diagnosis of COVID-19; many tests on the basis of S and N proteins have already been approved by the FDA.”	We have included as background the reasons for choosing NP and S proteins as the focus of the microarray, but we acknowledge in the discussion that inclusion of other proteins could improve performance based on new literature.
“More comprehensive SARS-CoV-2 protein microarrays have already been reported by other groups.”	We have cited additional literature as identified by the reviewers.
“It was unclear why the authors tested anti-IgA rather than anti-IgM.”	We have switched from IgA to IgM for detection of early infection in this revised manuscript.
“To assure the reliability of the array results, it is critical to assure the reproducibility of the newly fabricated microarray at the first instance.”	We show similar performance of an earlier iteration of the CoVAM in Figure 1 to demonstrate reproducibility of the assay across separate printing and probing runs.
“Because of the requirement of sophisticate expertise and special instruments, it is impractical to use protein microarray as a tool for routine diagnostics.”	We include as discussion a statement on use of portable cost-effective imaging modalities in analysis of protein microarrays.

Reviewers' Comments:

Reviewer #1:

Remarks to the Author:

1. There is no innovation or important findings in this study that worth the publication in an impact factor 12 journal as prestigious as Nat commun.
2. This paper only fits to go for a specialist journal such as Journal of Clinical Microbiology.

Reviewer #2:

Remarks to the Author:

Many thanks to the authors for updating the manuscript to address the comments.

The revision in the results section and the additional information in the discussion section are acknowledged.

There is no further comment.

Reviewer #3:

Remarks to the Author:

The authors have adequately addressed all of my concerns with additional data and reasoning.

Therefore, I feel the quality of the manuscript has improved significantly. I do not have any further concerns.